# A *Salmonella* Typhi Controlled Human Infection Study for Assessing Correlation between Bactericidal Antibodies and Protection against Infection Induced by Typhoid Vaccination

**DOI:** 10.3390/microorganisms9071394

**Published:** 2021-06-28

**Authors:** Elizabeth Jones, Celina Jin, Lisa Stockdale, Christina Dold, Andrew J. Pollard, Jennifer Hill

**Affiliations:** Oxford Vaccine Group, Department of Paediatrics, University of Oxford, and the NIHR Oxford Biomedical Research Centre, Oxford OX3 9DU, UK; celina.jin@paediatrics.ox.ac.uk (C.J.); lisa.stockdale@ndm.ox.ac.uk (L.S.); christina.dold@paediatrics.ox.ac.uk (C.D.); Andrew.pollard@paediatrics.ox.ac.uk (A.J.P.); jennifer.hill@ndm.ox.ac.uk (J.H.)

**Keywords:** Vi vaccination, SBA, luminescence, enteric fever, conjugate vaccine, correlates of protection

## Abstract

Vi-polysaccharide conjugate vaccines are efficacious against typhoid fever in children living in endemic settings, their recent deployment is a promising step in the control of typhoid fever. However, there is currently no accepted correlate of protection. IgG and IgA antibodies generated in response to Vi conjugate or Vi plain polysaccharide vaccination are important but there are no definitive protective titre thresholds. We adapted a luminescence-based serum bactericidal activity (SBA) for use with *S.* Typhi and assessed whether bactericidal antibodies induced by either Vi tetanus toxoid conjugate (Vi-TT) or Vi plain polysaccharide (Vi-PS) were associated with protection in a controlled human infection model of typhoid fever. Both Vi-PS and Vi-TT induced significant increase in SBA titre after 28 days (Vi-PS; *p* < 0.0001, Vi-TT; *p* = 0.003), however higher SBA titre at the point of challenge did not correlate with protection from infection or reduced symptom severity. We cannot eliminate the role of SBA as part of a multifactorial immune response which protects against infection, however, our results do not support a strong role for SBA as a mechanism of Vi vaccine mediated protection in the CHIM setting.

## 1. Introduction

Typhoid fever is a febrile illness caused by infection with the Gram-negative bacteria *Salmonella* enterica serovar Typhi. It is estimated to cause 14.3 million cases per year, leading to approximately 136,000 deaths [1,2,3]. Typhoid fever primarily affects low and lower-middle income countries and the major burden of disease lies in Asia and sub-Saharan Africa, where school age children are disproportionately affected [4]. Long term strategies to control enteric fever involve improvements in sanitation, infrastructure, and education, which can be costly and slow [5]. Effective vaccination programmes are important in helping to limit disease in the medium term until improvements in water quality are implemented, and are even more important given the emergence of antimicrobial resistant strains and outbreaks [6].

Until 2018 there were two vaccines licensed in many countries available for typhoid fever; the Vi capsular polysaccharide parenteral subunit vaccine, Vi-PS; and Ty21a, a live attenuated oral vaccine (Vivotif, Crucell Vaccines, Leiden, The Netherlands) [7]. These vaccines are not suitable for use in children under 2 or 6 years respectively, they offer only moderate efficacy and do not confer long term protection [8,9]. Protection following inoculation with purified Vi polysaccharide, a T cell-independent antigen, illustrates the key importance of Vi-specific humoral immunity for the control of typhoid fever. Recently a new conjugate typhoid vaccine has been developed which links Vi polysaccharide to a carrier protein to stimulate T cell help, driving immunological memory. The Vi tetanus toxoid conjugate vaccine (Vi-TT, Typbar-TCV, Bharat Biotech, Hyderabad, India) has been shown to be safe and efficacious in both adults and children [10,11,12]. The efficacy of the Vi-TT vaccine was demonstrated to be similar to the Vi-PS vaccine in healthy adults in a controlled human infection model (CHIM) [10]. Early results from a Phase III trial in Nepal demonstrate a protective efficacy of 81.6% after one year in children between 9 months and 16 years [11]. Therefore, Vi-TT vaccine and other Vi conjugates are promising candidates for the future control of typhoid fever, even in young children.

Currently, there is no accepted correlate of protection associated with either natural exposure, or vaccine induced protection for typhoid fever. Defining a correlate of protection could allow licensure of next generation vaccines without costly large scale efficacy trials, and would provide insight into the mechanism of action of these vaccines [13]. Knowledge of the mechanisms driving protection from *S.* Typhi infection may also contribute to the understanding of immunity to other typhoidal *Salmonella* serovars such as *Salmonella* Paratyphi A, the second leading cause of enteric fever, for which there is currently no licensed vaccine [3].

We have previously shown in a CHIM study where participants were allocated to receive Vi-TT, Vi-PS or a control vaccine (MenACWY) prior to oral challenge with 1–5 × 10^4^ colony forming units (CFU) *Salmonella* Typhi (Quailes strain), individuals with higher anti-Vi IgG titres at the time of exposure were less likely to develop acute typhoid fever, yet IgG antibody titre alone was a relatively poor indicator of protection [10]. Vi IgA quantity and fold change after vaccination were strongly associated with outcome of challenge [14]. The capacity of Vi-specific antibodies to induce certain effector functions may more strongly associate with protection. We have previously observed that distinct protective signatures are induced by Vi-PS and Vi-TT vaccines [15]. These protective signatures comprise both quantitative antibody (IgG and IgA) as well as functional responses mediated by innate cells, however, these analyses did not include evaluation of serum bactericidal activity (SBA).

SBA assays measure in vitro serum antibody-mediated classical complement pathway activation and bacterial killing. antibodies have become a widely accepted correlate of protection for *N. meningitidis*, and have been used to assess and license new meningococcal vaccines [16]. An inverse trend has been reported for *S.* Typhi cases in Kathmandu, Nepal, where SBA titres increase with age, as typhoid fever incidence decreases [17]. Participants diagnosed with acute typhoid fever in a previous CHIM study were observed to produce high SBA titres, suggesting that disease and not merely exposure to *S.* Typhi induces serum bactericidal antibodies [18]. While acute disease generates a significant increase in SBA titre, a single exposure to *S.* Typhi in the CHIM did not significantly alter Vi-specific antibody levels, thereby potentially excluding Vi-specific antibodies as a major driver of this bactericidal activity [19]. Other CHIM studies have evaluated SBA following oral typhoid vaccination with Ty21a, and vaccine candidate M01ZH09. After one dose of M01ZH09 SBA titres were significantly increased. Higher SBA titres after vaccination with Ty21a or M01ZH09 correlated with reduced disease severity, however SBA titres were not significantly associated with protection against infection. SBA activity in this study was shown to be dependent on anti-LPS antibodies [18].

In summary the contribution of complement-dependent antibody bactericidal activity in controlling typhoid infection is not well understood, and the SBA response to Vi vaccines in humans has not been evaluated within a CHIM. Here we describe the use of a luminescence based SBA assay (L-SBA) for evaluation of responses after Vi-TT or Vi-PS vaccination and the association with outcome following *S.* Typhi challenge.

## 2. Materials and Methods

### 2.1. Sample Collection

Serum was collected from healthy adult volunteers with no history of typhoid infection or residency in a typhoid endemic area who were recruited to the phase 2b study detailed in Jin et al. [10], Clinicaltrials.gov NCT02324751. Individuals were randomised 1:1:1 to receive either a Vi-conjugate (Vi-TT; Typbar-TCV, Bharat Biotech, Hyderabad, India), Vi-polysaccharide (Vi-PS; TYPHIM Vi, Sanofi Pasteur, Lyon, France), or meningococcal ACWY-CRM conjugate vaccine (control; MENVEO, GlaxoSmithKline, Sovicille, Italy). Twenty-eight days after vaccination, participants underwent oral challenge with 1–5 × 10^4^ CFU of *S.* Typhi, Quailes strain. Participants attended follow up visits for 14 consecutive days, during which time they could be diagnosed with typhoid fever by having a positive blood culture (daily samples taken for culture using the Bactec System, BD) or a fever greater than 38 °C for 12 h, before commencing antibiotics. Serum samples for SBA analysis were collected immediately prior to vaccination (pre-vac, PV) and 28 days later at time of *S.* Typhi challenge (D28).

For the Vi antibody depletion analysis, samples were taken from healthy adult volunteers who received either Vi-TT (*n* = 3) or Vi-PS (*n* = 4) but were not challenged with *S.* Typhi. Samples were collected immediately prior to vaccination (pre-vac, PV) and 4–6 weeks later (post-vac).

### 2.2. Luminescent-Serum Bactericidal Activity Assay (L-SBA)

The L-SBA assay methods are based on the publication by Necchi et al. [20]. Briefly, participant serum samples were heat-inactivated by incubating in a water bath at 56 °C for 30 min before making a dilution series starting at 1:1.5 in PBS. *S.* Typhi bacteria (Quailes strain, a wild-type, Vi+ strain isolated from a chronic carrier) were grown to log phase and diluted 1:60 in LB broth (Sigma, St. Louis, MO, USA) before adding to diluted test sera in the presence of 10% rabbit complement (CedarLane, Burlington, Canada) (10 µL diluted bacterial suspension, 10 µL test serum, 10µL rabbit complement, 70 µL LB) in a 96 well plate (VWR Ltd., Radnor, PA, USA). Plates were incubated for 3 h at 37 °C, with shaking at 220 rpm before centrifugation at 3220× *g* for 10 min. The supernatant was discarded and the pelleted bacteria re-suspended in PBS. The resultant bacterial suspension was transferred to a white flat-bottomed 96 well plate (VWR International Ltd., Radnor, PA, USA) and mixed in a 1:1 ratio with Promega BacTiter-Glo for quantification of luminescence (relative light units, RLU) using the LUMIstar OMEGA (BMG Labtech, Ortenberg, Germany), RLU output is directly proportional to the number of whole bacteria in the final suspension. SBA titres were calculated by normalising the luminescence measured for each sample dilution by that of the active complement only control (no serum), before fitting a 4-parameter sigmoidal curve to each dilution series and determining the serum dilution at which 50% killing of *S.* Typhi occurred. Data were included only if the R2 of the sample dilution curve was greater than 0.7, and the sample had been sufficiently diluted so the final RLU was comparable to the RLU of the complement only control. Samples were re-run if the titre of the positive control sample (international Vi standard 16/138, NIBSC UK) run on the same plate fell out of range (average ± 1 standard deviation). The limit of detection was defined as an SBA titre of 39.

### 2.3. Antibody Depletions

For depletion of Vi antibody, 96 well plates (Nunc maxisorp) were pre-coated for two hours with 10 ug/mL of poly-L-Lysine (Sigma, St Louis, MO, USA) before coating with 20 ug/mL *Citrobacter freundii* Vi polysaccharide (NIBSC, 12/244) [21]. Plates were washed with buffer containing 0.85% NaCl (Sigma, St. Louis, MO, USA) and 0.1% Brij 35 (Sigma, St. Louis, MO, USA), serum samples were added to wells in the first column of the plate, and the plates were sealed to avoid drying out. After 30 min at 37 °C, serum was carefully transferred into wells in the second column, avoiding contact with the bottom of the well. This was repeated for 9 further cycles. Antibody titres were then measured using the Binding Site VaccZyme *Salmonella* Vi ELISA kit, as per the manufacturer’s instructions [10].

### 2.4. Collection of Correlation Variables

Multiple clinical observations were measured and described by Jin et al. [10]. Quantification of anti-Vi antibodies and effector functions have also been measured previously, as detailed Jin et al. [15]. Here we use the previously collected data to investigate the relationship between SBA and various antibody properties, or clinical presentation of acute typhoid fever.

### 2.5. Statistical Analysis

Statistical analyses were performed using GraphPad Prism Version 8.3.0. Between timepoint comparisons were tested for significance using Wilcoxon matched pairs signed-rank tests and between group comparisons using Mann Whitney tests. Correlations were assessed using Spearman r test. *p* values ≤ 0.05 are considered statistically significant, no corrections were made for multiple testing. Sample size estimates were based on calculating vaccine efficacy.

## 3. Results

SBA titres were measured in samples at pre-vaccination and 28 days later at the point of *S.* Typhi challenge, a total of 72 vaccinated volunteers who completed the 14 day challenge follow up were included in the analysis, of whom 35 received Vi-PS and 37 received Vi-TT. Serum from a subset of six individuals who were vaccinated but not challenged were depleted of Vi antibody and tested for bactericidal activity pre-vaccination and post-vaccination (Vi-PS, *n* = 4. Vi-TT, *n* = 2).

### 3.1. Serum Bactericidal Activity Increases after Vi Vaccination

Serum bactericidal activity was evaluated prior to vaccination and 28 days post-vaccination in healthy volunteers who received either a Vi-PS or Vi-TT vaccine, before undergoing oral challenge with 1–5 × 10^4^ CFU wild-type *S.* Typhi. Both vaccines were associated with significant increases in SBA titre after 28 days (Vi-PS D28 titre median: 1848; CI 95% 1166-2464, *p* < 0.0001, Vi-TT D28 median 2001; CI 95% 1281-2923, *p* = 0.003) (Figure 1A and Appendix A). Though the difference in SBA between samples obtained prior to vaccination and at day 28 was greater in the Vi-PS recipients, there was no significant difference in SBA titres between the different vaccine groups at day 28 (*p* = 0.62). No significant differences were found when comparing fold change between PV and D28 between challenge outcome groups.

To assess whether bactericidal antibodies might be associated with protection against development of typhoid fever we compared SBA titres on the day of challenge between individuals who went on to develop acute typhoid fever with those who were not diagnosed throughout the 14 day challenge period. There was no significant difference in SBA titre between those who were diagnosed and those who were not diagnosed Vi-PS: *p* = 0.88, Vi-TT: *p* = 0.29 (Figure 1B), nor between those who remained healthy and those who developed disease when both vaccine arms were combined (Appendix A, TD vs nTD: *p* = 0.48). There was no significant difference in fold change from pre-vac to day 28 between any groups, although there was a trend towards significance comparing fold change in Vi-PS nTD group (*n* = 22) with Vi-PS TD group (*n* = 13), *p* = 0.09).

### 3.2. Post Vaccination Serum Bactericidal Activity Correlates Weakly with Anti-Vi IgG, IgG2 and IgM Titres

SBA titres at day 28 post-vaccination were compared with other parameters to determine if vaccine-induced bactericidal antibodies were correlated with Vi-specific antibody levels (Figure 2) or with measures of diagnosis outcome (Table 1). SBA titre 28 days after Vi-PS vaccination weakly correlated with anti-Vi IgG, IgG2, and IgM at this time point, and this was driven by individuals who did not develop typhoid fever (nTD) (Figure 2A–C). In contrast, SBA titres 28 days after Vi-TT vaccination did not significantly correlate with Vi-IgG, Vi-IgG2 or Vi-IgM.

SBA titre at the point of challenge did not significantly correlate with Vi-specific antibody-mediated complement deposition, nor with any clinical parameters relating to disease severity (Table 1). 

### 3.3. Depletion of Vi Antibodies Reduces Serum Bactericidal Activity

To demonstrate that Vi-TT/Vi-PS vaccine-induced SBA is mediated by Vi-specific antibodies, Vi-specific antibodies were depleted in samples from a subset of individuals (*n* = 6, individuals for this subset were chosen due to availability of large serum volumes). Anti-Vi IgG was significantly reduced following depletion in pre-vac and day 28 samples (Figure 3A). Geometric Mean Titre (GMT) pre-vaccination prior to depletion was 45.8 EU/mL which reduced to 17.5 EU/mL following depletion, while for day 28 the GMT prior to depletion was 669.1 EU/mL and 106.4 EU/mL following depletion. Analysis of SBA in these samples shows a corresponding significant reduction in activity after Vi antibody depletion at day 28 only with titres comparable to pre-vaccination levels (Figure 3B; *p* = 0.56). SBA GMT pre-vaccination versus day 28; 1190 to 2002 originally, 809.2 to 1232 in depleted samples).

## 4. Discussion

Here we describe and compare induction of bactericidal antibodies after vaccination with two Vi-containing vaccines and explore the relationship between SBA and protection against acute typhoid fever in a controlled human infection model (CHIM). We show that Vi vaccination induces a significant increase in bactericidal antibodies after 28 days but find no evidence these are a correlate of protection within a CHIM setting.

Consistent with literature describing other *S.* Typhi vaccines in human and animals, such as live attenuated oral vaccine candidates M01ZH09 and CVD910, or intramuscular Vi-purified capsular polysaccharide, we show that vaccination with either Vi-PS or Vi-TT leads to significant increases in SBA titre [22,23,24]. We did not detect a difference in SBA titre between individuals who developed acute typhoid fever and those who did not after deliberate human challenge one month after vaccination, nor a significant association with disease severity (Figure 1, Table 1) which has been previously described after oral typhoid vaccine. This also contrasts with other enteric disease such as cholera, where SBA is a correlate of vaccine-induced protection, and shigellosis where vaccine mediated SBA is associated with reduced disease severity in a controlled human infection model [25,26].

Induction of high titres of bactericidal antibodies after a single dose of oral vaccine candidate M01ZH09 has been shown to reduce the severity of disease within the context of the *S.* Typhi CHIM [18,23]. Studies in typhoid endemic areas have also demonstrated that anti-Vi titres and SBA titres correlate with the age distribution of disease burden, though the exact relationship between antibodies and protection to natural infection is unknown [17]. However, there was no significant correlation between bactericidal antibodies and anti-Vi titres, suggesting the relationship between SBA titre and disease burden is at least in part due to other antibodies targeting other antigens.

One explanation for the differences observed in the relationship between SBA and disease severity when comparing post vaccine responses from M0IZH09 and Vi vaccines could be due to the due to the specificity of the antibodies that the vaccines elicit. Vaccine candidate M0IZH09 induces strong LPS specific antibodies, whereas Vi vaccination antibody responses are limited to Vi. It has been hypothesised that expression of Vi capsular polysaccharide by *Salmonella* bacteria is a virulence mechanism helping bacteria avoid the immune system by reducing complement mediated killing [27]. However, in vitro experiments show that anti-Vi antibodies can bring about killing of Vi-expressing bacteria in the presence of complement [28]. To determine the specificity of the bactericidal antibodies present in our volunteers at both baseline and after Vi vaccination, we carried out the SBA assay using samples depleted of Vi antibodies. Depleted samples showed significantly reduced bacterial killing compared with non-depleted samples, and depletion reduced post-vaccine SBA titres to baseline levels, confirming that vaccine induced SBA was Vi-mediated rather than being caused by antibodies generated against other, non-Vi antigens. Interestingly, we observed a range of SBA titres at baseline, presumably partially due to pre-existing Vi antibodies from possible unknown exposure to *Salmonella* species or other bacteria such as *Citrobacter freundii*; a commensal commonly present in the gut which expresses Vi antigen [21]. Some pre-existing bactericidal activity might also be attributable to antibodies against other target antigens such as LPS which is abundant in the microbiome, or other antigen homologs. It is possible that antibodies against LPS have more potent SBA in vivo, potentially explaining why we see a correlation with reduced severity of disease with live attenuated vaccines which can induce LPS antibodies, and do not in the case of Vi vaccines which contain minimal amounts of contaminant LPS. While some data suggest that long chain LPS has a similar virulence mechanism as Vi polysaccharide, functional antibodies made against these antigens may remain quite distinct [29].

Another important factor to consider when comparing observations from multiple studies is the difference in methods. Each of the studies described uses different methods to assess bactericidal activity which could account for the observed differences, the sensitivity of the bacteria could be different depending on culture conditions, as seen in Boyd et al. [22] who describe a resistance to killing of stationary phase *S.* Typhi compared with log phase bacteria. Antigenic expression could also vary between culture methods and might be responsible for the variation in relationship between bactericidal antibodies and protection from infection or disease severity. It is thought that as little as 2% of the Vi capsule remains attached to the surface of the bacteria in vitro [21]. Capsule fractions that have sloughed off the surface may still remain intact in the culture and bind antibody in the serum reducing the pool available for the bactericidal activity, this may explain why we see relatively little increase in SBA titre post-vaccination while the corresponding increase in Vi antibody titres is much greater [10].

It is possible that Vi-mediated SBA is not a strong correlate within our model due to variable expression of the Vi capsule within an individual during the course of infection, as we know a single exposure to *S.* Typhi in a CHIM is not enough to cause an increase in Vi antibody levels [19]. Expression of the Vi capsule is highly regulated by genes located on *Salmonella* pathogenicity island 7 (SPI-7), and is thought to be expressed under low osmolarity conditions as the bacteria translocate the intestinal epithelial barrier where bactericidal activity is not thought to occur due to lack of complement and complement fixing antibodies [21]. Capsule expression during other stages of the infection process are not well defined and while it’s thought expression is low during the intracellular phase little is known about expression by extracellular bacteria. This, along with the facultative intracellular nature of *S.* Typhi, mean that while the antibodies are present, their target antigen is not readily seen and recognised, therefore antibodies are unable to effectively kill the bacteria.

Correlations of SBA titres at the point of *S.* Typhi challenge with antibody measures such as isotype and subclass titres, and propensity to mediate complement deposition showed that among Vi-PS recipients, SBA titre significantly correlated with anti-Vi IgG, IgG2 and IgM. This finding is expected as IgM is the key isotype for complement mediated killing, due to its pentameric structure and capacity for multiple epitope binding, and IgG2 being the main immunoglobulin produced in response to plain polysaccharide antigens (Figure 2, Table 1) [30,31]. No significant correlations between SBA titres and other antibody measures were found in individuals who received the Vi-TT vaccine (Figure 2, Table 1). Interestingly no correlation was found between SBA titre and antibody dependent complement deposition (ADCD). Differences in methods may partially account for this, ADCD used Vi coated beads rather than the live bacteria used in the SBA assay. ADCD only measures C3b deposition, rather than membrane attack complex (MAC) formation and bacterial lysis quantified in the SBA assay. Our SBA used rabbit serum as complement source whereas the deposition assay used guinea pig serum, variations in the potencies of individual complement pathway components between these sources could explain some of the differences observed. Thus, the lack of correlation between SBA and ADCD could be due to inherent differences in complement, or effectors downstream of the C3b stage, or variable Vi expression of the bacteria. Neither bactericidal activity or complement deposition correlate highly with protection within our challenge model [15].

We also investigated the relationship between SBA and clinical and microbiological measures of disease including bacterial load and duration of fever, none of which showed any significant relationship for either vaccine arm (Table 1). Previously anti-LPS mediated SBA has been shown to correlate with reduced disease severity, this highlights the different mechanisms by which vaccines mediate protection as well as potential differences in antigen specific antibody effector functions. We conclude that Vi specific bactericidal antibodies have no obvious role in mitigating infection or disease severity within our model. However, since all volunteers are treated by 14 days with antibiotics in our model we cannot determine whether bactericidal antibodies may impact long term transmission or development of the chronic carrier state in endemic settings.

The sample sizes in this study were calculated to investigate the protective efficacy of 2 Vi vaccines compared with control vaccine MenACWY, the study was not powered to determine a correlate of protection. Subtle changes in SBA titre after vaccination (average fold change 0.96) and evidence of a large range of pre-existing bactericidal titres may hinder our ability of identifying a correlation between bactericidal activity and protection. Our ability to reliably identify a relationship between SBA and protection or disease severity is further hindered by such small numbers of diagnosed individuals (Vi-PS; TD *n* = 13, Vi-TT; TD *n* = 13).

Previous analysis of biophysical and functional antibody properties in our study cohort show that anti-Vi IgA quantity and avidity are strongly associated with protection after Vi vaccination [14,15]. However, IgA is not known to mediate complement dependent killing via that classical pathway as it is a poor activator of C1q that lacks a binding site in the FC region [32], nor does it correlate with SBA in our analyses. Although we cannot rule out that SBA-inducing antibodies may be working alongside other functions to protect against infection, our results do not support a strong role for SBA as a mechanism of Vi vaccine mediated protection in the CHIM setting. Added to our existing knowledge, these results demonstrate the varied mechanisms of action of different types of vaccines, and highlight that protective correlates may differ between vaccines and may be multifactorial rather than being mediated by a single function as explained by Plotkin et al. [33].

## Figures and Tables

**Figure 1 microorganisms-09-01394-f001:**
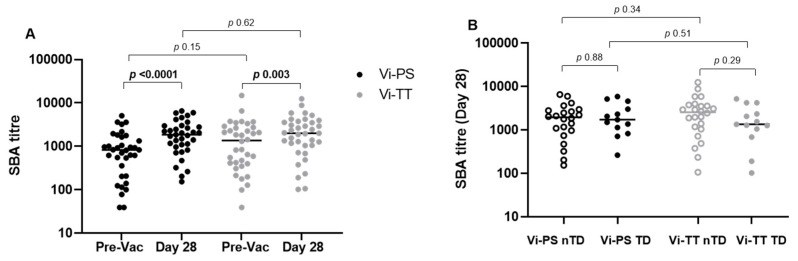
Serum bactericidal activity was evaluated prior to vaccination and 28 days post-vaccination in healthy volunteers who received either a Vi-PS or Vi-TT vaccine, before undergoing oral challenge with 1–5 × 10^4^ CFU wild-type *S.* Typhi (**A**) Comparisons between time points, split by vaccine Vi-PS *n* = 35, Vi-TT *n* = 37. (**B**) Comparisons of day 28 titres, split by vaccine and challenge outcome (TD; typhoid diagnosed, closed circles. nTD; not typhoid diagnosed, open circles). Limit of detection is an SBA titre of 39. Between time point comparisons analysed using Wilcoxon matched-pairs signed rank test, between group comparisons analysed by Mann-Whitney.

**Figure 2 microorganisms-09-01394-f002:**
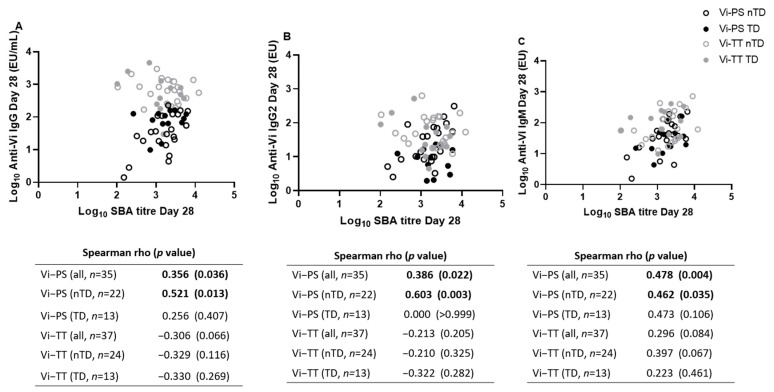
Correlation between anti-Vi IgG, IgG2, and IgM levels and SBA titre at the time of challenge. Correlations between SBA titre at the point of challenge with anti-Vi IgG (**A**), anti-Vi-IgG2 (**B**), and anti-Vi-IgM (**C**). TD; typhoid diagnosed, closed circles. nTD; not typhoid diagnosed/remained well, open circles.

**Figure 3 microorganisms-09-01394-f003:**
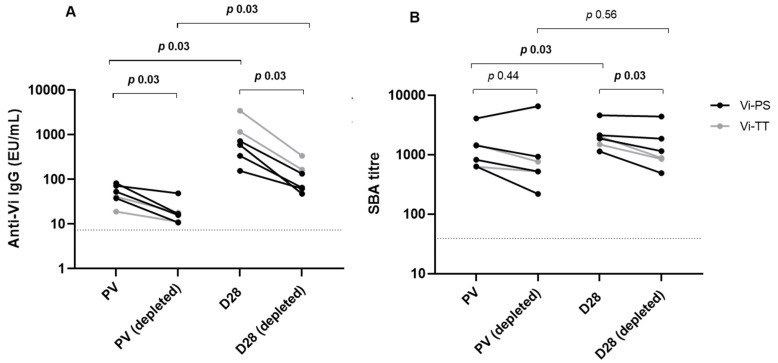
(**A**) Anti-Vi IgG titre determined by ELISA, pre and post Vi antibody depletion, in a subset of Vi-PS and Vi-TT vaccinated individuals. Dotted line marks the limit of detection, 7.4 EU/mL. (**B**) Corresponding SBA titres from pre and post depleted serum. Dotted line marks the limit of detection, 39. (*n* = 6).

**Table 1 microorganisms-09-01394-t001:** Table of correlation outcomes of log_10_ SBA titre 28 days after vaccination compared with various parameters. Anti-Vi antibody levels and antibody dependent complement deposition data all 28 days post vaccination, at the point of challenge.

Spearman Rho (*p* Value)						
	Vi−PS (all, *n* = 35)	Vi−PS (nTD, *n* = 22)	Vi−PS (TD, *n* = 13)	Vi−TT (all, *n* = 37)	Vi−TT (nTD, *n* = 24)	Vi−TT (TD, *n* = 13)
Anti−Vi IgG1	0.226 (0.192)	0.261 (0.241)	0.102 (0.740)	−0.152 (0.371)	−0.040 (0.845)	−0.382 (0.197)
Anti−Vi IgG3	0.230 (0.183)	0.245 (0.272)	0.280 (0.354)	−0.070 (0.682)	0.044 (0.837)	−0.429 (0.144)
Anti−Vi IgA	0.079 (0.657)	0.286 (0.205)	−0.223 (0.461)	−0.092 (0.593)	−0.232 (0.287)	−0.052 (0.867)
Antibody Dependant Complement Deposition	0.178 (0.307)	0.350 (0.110)	−0.107 (0.763)	0.051 (0.763)	−0.005 (0.982)	0.070 (0.818)
Time to Typhoid Fever Diagnosis (Days)	0.064 (0.716)	NA	0.252 (0.404)	−0.218 (0.196)	NA	−0.209 (0.490)
Duration of Fever (Days)	0.089 (0.617)	0.333 (0.140)	−0.772 (0.365)	−0.038 (0.825)	−0.243 (0.253)	0.333 (0.262)
Duration of Bacteraemia (Days)	−0.062 (0.731)	NA	−0.182 (0.589)	−0.178 (0.298)	NA	0.470 (0.125)
Symptom Severity	0.133 (0.4454)	0.132 (0.558)	0.011 (0.978)	−0.266 (0.112)	−0.312 (0.138)	−0.124 (0.684)
Bacterial Burden (CFU/mL)	0.127 (0.733)	NA	0.127 (0.733)	0.112 (0.760)	NA	0.112 (0.759)

## Data Availability

The datasets generated for this study are available on request to the corresponding author.

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
