# Peer review of "A Salmonella Typhi Controlled Human Infection Study for Assessing Correlation between Bactericidal Antibodies and Protection against Infection Induced by Typhoid Vaccination"

_microorganisms, 2021, doi:10.3390/microorganisms9071394_

Round 1
Reviewer 1 Report
This paper shows an important study assessing correlation between bactericidal antibodies induced by Typhoid Vaccination and protection against infection in a controlled human Salmonella Typhi infection setting.
Data do not support a statistical correlation between bactericidal antibodies and protection.
1) Pre-existing bactericidal antibodies (prior to vaccination) seem already relatively high and, although significantly different, there is not a large increase after vaccination. In addition, there is quite large variability in titers both prior and after vaccination. These factors may hinder the ability of identifying a correlation between bactericidal activity and protection. They authors should add this aspect in their discussion as a potential limitation of the study.
2) Another limitation that the authors should highlight is the relatively small sample size available in the study for comparing volunteers who developed the disease and those who did not. It should also be emphasized that the small sample size in association with the pre-existing immunity and the wide range of the titers further limits the ability to reliably interpret the data.
3) The authors mention a potential important role played by IgA. The authors should expand this aspect, attempt to perform experiments assessing this hypothesis or explain why this could not be assessed.
4) The title of the manuscript seems not to properly reflect the findings of the study. I believe that this study does not show that bactericidal antibodies do not mediate protection. I think that this study is better reflected by the following titles: “A Salmonella Typhi Controlled Human Infection study for assessing correlation between Bactericidal Antibodies and protection against infection Induced by Typhoid Vaccination”.
Author Response
Thank you for your comments and feedback
1 & 2. These points have been addressed together is a new paragraph in the discussion (lines 369-376).
3. I think IgA is known to not play an important role in the classical complement cascade as it is a poor C1q activator, therefore the relationship between IgA and SBA is weak (no significant correlation demonstrated in table 1). The relationship between Vi IgA and protection is already a complete body of work described in reference 15, this has been clarified in the final paragraph.
4. Title has been changed as suggested.
Thank you for you helpful comments and feedback, I hope that I have addressed them appropriately
Reviewer 2 Report
Introduction
- Please review the name of microorganisms in text. Some of them (line 74,76) are not in italic.
- The introduction is very complete, but it could be shorter. Some of the information can used in the discussion instead, and some paragraphs are long (line 72 to 92). I suggest 5 paragraphs maximum (8 – 10 lines/paragraph)
Results
- The figure 1B is difficult to understand at first look. Also, the legend is not clear either. However, in the text I could understand well. I don’t see how to change the figure, but I think you can modify the legend. Maybe adding something like what is described in line 187 to 190.
- Do the authors have kinetic data (as supplementary data) supporting 28 days post-vaccination as optimal time point for SBA titres?
- What is sensitivity of SBA test? What is the limit of detection?
- Figure 2 is too small. Can the author make a large figure instead?
- The table 1 seems to be made in gray, and the letters/numbers are very small. Can the authors make a large table as well?
- The reduction of SBA in the depletion system is hard do believe. Any comments? Why is the SBA so high in the PV group?
- Have the authors tried to purify the anti-Vi antibodies (IgG, or IgA) and use them in SBA?
- Can the author run SBA with D28 serum IgA-depleted, IgG-depleted, or IgM-depleted? It difficult to see the contribution of each class in the BSA. Maybe this approach combined with suggestion 7 can provide some interesting results/insights.
- A good control for figure 3 could be a serum completely depleted of antibodies.
- Where is the control group vaccinated with meningococcal conjugate vaccine in the figures?
- Do the authors have any piece of data about bactericidal antibodies against S.Typhi and disease severity?
Discussion
- It’s a solid discussion. I suggest avoid long paragraphs (line 259-281;309-327).
Author Response
Thank you for your comments and suggestions.
- Names of microrganisms changed to correct italics - thank you for picking this up
- Paragraph 5 of introduction edited to shorten it
- No kinetic data was collected, main aim of the work was to look at SBA as a correlate of protection, D28 was picked for that reason.
- The sensitivity limit is an SBA titre of 39. This has been added to the L-SBA section in the methods and the figure legends for fig 1 and 3.
- Figure 2 has been enlarged slightly. The figures can be stacked vertically if they are still difficult to read at the size.
- Font size of the table increased to 12, font is in black but I think the small size was making it look grey.
- While the depletion in Vi antibody titre is strong the corresponding reduction in SBA is subtle, but it is statistically significant. Perhaps this demonstrates a redundancy of Vi antibody mediated SBA, i.e. other antibodies have space to bind to their antigens and therefore the reduction in SBA is not of the same magnitude as the antibody reduction. This effect may be partially attributable to low numbers (due to limited serum volumes). High PV SBA titres can be explained due to some baseline Vi titres (as seen in the ELISA data) or because of antibodies to other common antigens (such as LPS). This has been clarified in the discussion lines 297-300.
- No attempts have been made to purify Vi specific antibodies
- I think it would be really interesting to either purify Vi specific antibodies or investigate the effect of total depletion of IgG/M on SBA. However this is a very serum thirsty method and we have limited volumes left, so I think we wouldn't end up with a good n.
- For each sample, we dilute it out so it provides a similar level of bacteria survival as a complement only control which is potentially acting as a similar control here. I think without antibodies we wouldn't be able to achieve a killing curve and therefore could not calculate an SBA titre so it would not be a graphable control.
- Meningococcal vaccine control participant samples were not run. The main aim of this work was to investigate SBA as a correlate of protection after Vi vaccine, given the recent licensure of Vi conjugate vaccines.
- the relationship between SBA and disease severity if investigated in table 2, and described in the results section lines 215-217. Comparisons between this observation and historic data (collected from a CHIM and a study in Nepal) are discussed in the discussion in paragraph 3 & 4.
- Paragraph 5 (lines 309-327) edited to shorten. Lines 259-281 appearing as 3 separate short paragraphs.
Round 2
Reviewer 2 Report
No comments